# Geospatial Heterogeneity in Monetary Value of Proximity to Waterfront Ecosystem Services in the Gulf of Mexico

**Ram P. Dahal [1,\*], Robert K. Grala [2], Jason S. Gordon [3], Ian A. Munn [2] and Daniel R. Petrolia [4]**

[1] Wisconsin Department of Natural Resources, PO Box 7921, Madison, WI 53707-7921, USA

[2] Department of Forestry, Mississippi State University, Box 9681, Starkville, MS 39762, USA; r.grala@msstate.edu (R.K.G.); iam1@msstate.edu (I.A.M.)

[3] Warnell School of Forestry and Natural Resources, University of Georgia, Athens, GA 30602, USA; jason.gordon@uga.edu

[4] Department of Agricultural Economics, Mississippi State University, Box 5187, Starkville, MS 39762, USA; d.petrolia@msstate.edu

\* Correspondence: ram.dahal@wisconsin.gov

**Abstract:** Open spaces, including waterfront areas, are critical to coastal communities and provide many benefits, including recreation opportunities, economic development, ecological benefits, and other ecosystem services. However, it is not clear how values of waterfront ecosystem services vary across geographical areas which prevents development and adoption of site-specific natural resource conservation plans and suitable long-term land management strategies. This study estimated the monetary value of distance to different waterfront types in coastal counties of Mississippi and Alabama (U.S.) using a geographically weighted regression (GWR) approach as an extension to a traditional hedonic pricing method (HPM). In addition, the study utilized publicly available data from the U.S. Census Bureau instead of certified rolls of county property assessors and Multiple Listing Service (MLS) data which can be costly and difficult to obtain. Residents valued most waterfront types which was reflected in greater assessed prices for houses in proximity to these waterfronts. However, the value of ecosystem services associated with waterfronts differed geospatially. The marginal implicit prices ranged from −$6343 to $6773 per km depending on a waterfront type. These estimates will be useful to city developers, land-use planners, and other stakeholders to make more informed and balanced decisions related to natural resource preservation associated with coastal areas, land-use planning, and zoning. In addition, information from this study can be used in developing healthy living environments where local economy can benefit from increased property tax revenues associated with waterfronts and their ecosystem services.

**Keywords:** geographically weighted regression; hedonic pricing method; environmental amenities; spatial variability; marginal implicit price; open space

## 1. Introduction

As coastal areas experience rapid population growth, waterfront open space is increasingly limited resulting in overuse and a decreased potential to provide ecological, sociocultural, and economic benefits [1,2]. Urban waterfronts represent publicly or privately-owned landscapes that are part of a city and in contact with a water body such as an ocean, lake, river, or estuary [3–6]. These areas provide many benefits and ecosystem services such as natural scenery; ecological benefits such as wildlife habitat and urban heat reduction; recreational opportunities such as hiking and boating; and produce economic benefits related to higher real estate values and increased tax revenues [7–10]. Waterfronts are therefore important to coastal communities and their visitors, and future land-use planning decisions should balance their preservation and development in ways

that enhance cultural, environmental, and economic values of open spaces and adjacent areas. Considering the distribution of waterfronts across the community landscape, including its socioeconomic geography, can identify ways to improve resident access to open space ecosystem services.

Population growth is one of the major threats to coastal waterfront open spaces [11,12]. For instance, the U.S. coastal population, including Atlantic, Gulf of Mexico, and Pacific, grew by 40 million between 1960 and 2008, representing an increase of 84% [13]. During the same time period, the U.S. population in the coastal counties along the Gulf of Mexico increased by 150%, whereas in five coastal counties in Mississippi and Alabama, including Hancock, Harrison, Jackson, Mobile, and Baldwin, population increased by over 65% taking into account a population decline after numerous hurricanes such as Katrina [13]. Similarly, between 2010 and 2019, population in these five counties increased by another 8% overall, of which Baldwin County experienced the largest increase of over 20% [14]. Availability of open space, such as waterfront open space, is particularly important in urban environments where a rapid population increase and land scarcity can lead to potential land-use conflicts [15]. Thus, preservation of waterfront open space versus its conversion to residential and commercial uses has become an important policy issue in land-use planning and urban development.

Ecosystem services and goods provided by waterfronts are important to human welfare; however, they are often neglected in land-use decisions because many of them do not provide direct monetary remuneration, potentially leading to their suboptimal provision. Monetary valuation of such ecosystem services enables quantification of economic tradeoffs between preserving waterfront open space versus development (with associated property taxes) to better inform future urban development decisions [6,16]. Many previous studies have quantified a monetary value of various nonmarket ecosystem services provided by waterfront open spaces using a hedonic pricing method (HPM). For example, Gibbons et al. [17] estimated that a 1 km increase in distance to river and coastline decreased the house price by 0.93% and 0.14%, respectively. Scorse et al. [18] did a similar study to quantify monetary value of proximity to oceanfront in California, U.S. and estimated that houses located next to the surf break were valued $106,000 more than similar houses located one mile away. Authors also determined that a house with a full ocean view was valued almost $1 million more than a similar house with no ocean view. Another study by Anderson and West [19] estimated an increase in house value ranging from 0.004% to 0.034% with a 1-percent decrease in distance to neighborhood parks, special parks, lakes, and rivers. Bin [20] reported that moving 300 m closer to the nearest river increased the property value by $3700, whereas Costanza et al. [21] demonstrated that houses within 90 m from a beach were sold for $81,000 to $194,000 more than those located farther away. Similarly, Poudyal et al. [22] estimated increases in the house price by 0.016% with a 1% decrease in distance to a park. Thus, while many waterfront ecosystem services are not traded in the markets, they are valued by residents which has been reflected in prices of market goods such as houses.

There are also numerous studies conducted in the Gulf of Mexico. For instance, a study done by Dumm et al. [23] in Hillsborough County in Florida estimated that in general waterfront houses had a 7.20% price premium added compared to non-waterfront houses. Similarly, a study by Dahal et al. [24] in the cities of Daphne and Mobile adjacent to the Gulf of Mexico estimated that house prices increased by $2500 to $15,500 per km with increase in proximity to different waterfront types. In another study, Dahal et al. [25] conducted a survey in four cities along the Gulf of Mexico to evaluate residents' attitudes towards waterfront open space. The study suggested that over 95% of respondents believed that it was important to preserve the coastal character of waterfronts in future development decisions. The study also revealed that a majority of respondents (71.29%) believed that commercial development and changing economy were major threats to the existence of waterfronts and more than half of the residents (53.96%) believed that these resources should be protected at any cost. Thus, waterfronts are important elements of

coastal communities and monetary valuation of ecosystem services provided by them can help identify priorities in their protection versus development.

The monetary value of nonmarket ecosystem services provided by waterfronts can be extrapolated from residential house sale transactions. While the HPM has been used extensively in nonmarket valuation, it typically relied on house values obtained from tax assessors or Multiple Listing Service (MLS) as a data source. However, obtaining MLS data, which is typically owned by a private MLS organization, or tax assessor data can be costly and time-consuming, and not readily available in every location. Furthermore, while house sale data can also be obtained from other sources, such as Zillow (one of the leading online real estate and rental marketplaces reporting information related to median house values, inventory, and other parameters), this approach may not be practical as decision-makers and practitioners are not necessarily aware of alternative sources and might not know how to incorporate such information into the land-use analysis. In addition, other sources might not be able to provide information with the resolution required for the analysis and their housing value estimates might not be as accurate as owner's estimates of property value [26]. Such difficulty in obtaining the required data can result in planners lacking important information when assessing the value of waterfront ecosystem services to guide land-use decisions and developing zoning guidelines.

Many previous valuation studies have considered the impact of waterfront proximity on house value; however, they did not account for the effect of spatial variation in the impact of waterfront ecosystem services on house values (e.g., [27–35]). For example, Cohen et al. [36] demonstrated that the ordinary least squares (OLS) regression method did not indicate the relationship between house price and proximity to waterbodies; however, the geographically weighted regression (GWR) method suggested that a 1-percent increase in distance to a waterbody was related to a house price decrease of 0.027%. Similarly, Pandit et al. [37] and Tapsuwan et al. [38] suggested the use of spatial models to account for the impact of location on house prices because spatial dependencies can result in biased and inconsistent estimates when they are derived from stationary models such as the OLS. Information obtained from the OLS model can only provide average estimates that can serve as a useful benchmark for making general statements; however, such estimates might not accurately reflect monetary values of ecosystem services associated with specific sites [39]. Thus, there is a need to account for spatial heterogeneity when quantifying a monetary value of ecosystem services associated with waterfronts to better understand how these attributes are related to house values and facilitate an improved planning for open space conservation and development. This study used a GWR model to determine spatial variation in monetary values of waterfronts adjacent to urban areas. Marginal implicit prices were calculated and then mapped to visualize the effect of waterfront's proximity on a house value. Findings from this study have several policy implications related to natural resource preservation associated with coastal areas, land-use planning, and zoning.

## 2. Materials and Methods

### 2.1. Study Area

The study was conducted in the five coastal counties in the Gulf of Mexico (U.S.), three from Mississippi and two from Alabama: Hancock, Harrison, Jackson, Mobile, and Baldwin (Figure 1). Total population in the study area in 2010 was approximately 950,000, of which Mobile accounted for 43.21%, Harrison for 19.22%, Baldwin for 18.59%, Jackson for 14.50%, and Hancock for 4.48% [14]. In terms of racial/ethnic composition, Mobile County had 61.30% white, 34.50% Black or African American, and 4.20% Other. Harrison County had 71.00% White, 21.80% Black or African American, and 7.20% Other (7.20%). Baldwin County had 86.20% White, 9.50% Black or African American, and 4.30% Other. Jackson County had 73.80% White, 21.70% Back or African American, and 4.50% Other.

Hancock County had 89.20% White, 7.00% Black or African American, and 3.80% Other [14]. Thus, more that 80% of population in the study area was white suggesting low racial variability.

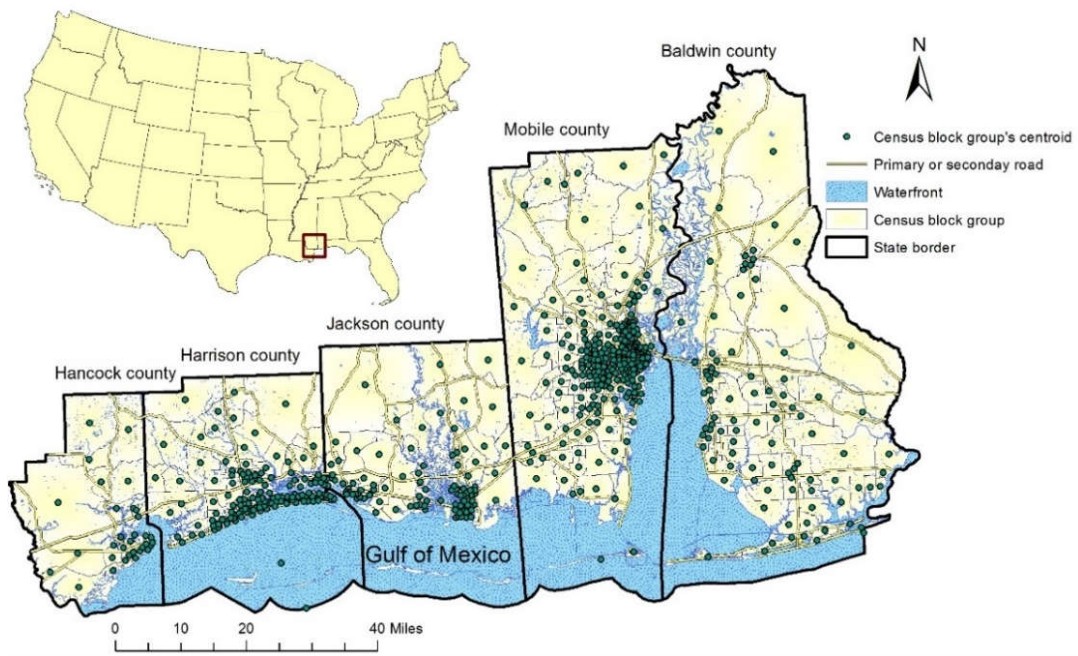

**Figure 1.** Location of the study area in the Gulf of Mexico.

According to the U.S. Census [40,41], 26.49% of the study area was covered by water, suggesting a considerable amount of waterfront open space. Of the total area, 30.70% was covered by wetlands, 25.21% by forest land, 12.46% by shrub land, 9.90% by planted or cultivated area, and 11.86% was developed. The remaining area (9.87%) included other land types such as barren and herbaceous land [42]. There are a number of popular destination beaches and several rivers that provide scenic and recreation amenities.

Despite the recession of 2001 and 2007–2009 and corresponding declines in the housing market, the number of housing units in the study area increased substantially for a net increase of 35.31% between 1990 and 2010 [40,41]. The area comprised of 633 Census block groups where each group represented a statistical division of a Census tract typically consisting of 600 to 3000 people [43]. Three block groups consisted of water only and were omitted from the analysis.

*2.2. Data Collection*

This study used Census data instead of MLS data. The Census data sets have several advantages because they are freely available to the public, spatially georeferenced via the Topologically Integrated Geographic Encoding and Referencing (TIGER) system and can be easily integrated with spatial data related to availability of ecosystem services using Geographic Information Systems (GIS). The use of aggregated Census data can serve as low cost and practical alternative data source for planners to estimate a monetary value of proximity to waterfronts and their ecosystem services.

Data related to house values and their structural attributes were based on the 2010 United States Census and were collected at Census block group level [44]. The Census block group is the smallest statistical unit with detailed house structural and neighborhood information required for modeling. Median house values (a dependent variable) represented Census respondents' estimates of how much the property would sell for in the current market and might differ from a price at which the property would

actually sell. Independent variables consisted of three sets of attributes: structural, neighborhood, and ecosystem services. Variables representing house structural attributes included number of rooms and house age. Neighborhood attributes included two groups of variables. The first group consisted of socioeconomic characteristics representing neighborhood development level, economic condition and prosperity such as household income, resident median age, percentage of families below poverty line, population density, and percentage of recreationally used houses and vacant houses, whereas the second variable group comprised of government and locational services represented by distance to the nearest facilities such as school, road, railroad, hospital, shopping center, and airport. The neighborhood data were obtained from various open sources such as the U.S. Census, Mississippi Automated Resource Information System (www.maris.state.ms.us, accessed on 20 June 2016), City of Mobile (www.cityofmobile.org, accessed on 20 June 2016), and expertGPS (www.expertGPS.com, accessed on 20 June 2016), whereas Euclidian distance from the centroid of the Census block group was estimated using Esri's ArcMap.

To identify ecosystem services, we followed Dahal et al. [25] in which the author conducted a detailed survey to determine how coastal communities valued and used waterfront open spaces in five coastal counties of Mississippi and Alabama. Following Dahal et al. [25], ecosystem services were represented by proximity to the nearest public park and different waterfront types. Data on location of ecosystem services were obtained from the U.S. Census Bureau [45]. The U.S. Census Bureau defined streams as natural flowing waterway with an intricate network of interlacing channels, artificial waterways constructed to transport water, to irrigate or drain land, and natural or artificial waterways to connect two or more bodies of water, or to serve as waterway for watercraft, whereas a river was defined as natural flowing waterway that did not contain the characteristic of a stream [46]. A bay was defined as a body of water partly surrounded by land and bayou as slowly moving water and marshy land [46]. Centroids of each Census block group were identified using Esri ArcMap and the Euclidian distance from each Census block group centroid and nearest waterfront was determined using ArcGIS. Table 1 provides descriptions of all variables used in the study.To identify ecosystem services, we followed Dahal et al. [25] in which the author conducted a detailed survey to determine how coastal communities valued and used waterfront open spaces in five coastal counties of Mississippi and Alabama. Following Dahal et al. [25], ecosystem services were represented by proximity to the nearest public park and different waterfront types. Data on location of ecosystem services was obtained from the U.S. Census Bureau [45]. The U.S. Census Bureau defined a stream as a natural flowing waterway with an intricate network of interlacing channels, artificial waterways constructed to transport water, to irrigate or drain land, and natural or artificial waterways to connect two or more bodies of water, or to serve as waterway for watercraft, whereas a river was defined as a natural flowing waterway that did not contain the characteristic of a stream [46]. A bay was defined as a body of water partly surrounded by land and bayou as slowly moving water and marshy land [46]. Centroids of each Census block group were identified using Esri's ArcMap and the Euclidian distance from each Census block group centroid and nearest waterfront was determined using Esri's ArcGIS. Table 1 provides descriptions of all variables used in the study.

**Table 1.** Definition of variables used in the study to estimate a monetary value of distance to waterfront ecosystem services in coastal counties of Mississippi and Alabama, U.S.

| Variable | Description | Mean | Std. Dev. |
|---|---|---|---|
| **Dependent Variable** | | | |
| House value (2010 US$ in thousands) | Median house assessed value as reported by residents in 2010 | 139.20 | 72.02 |
| **Independent Variables** | | | |

| House structural attributes | | | |
|---|---|---|---|
| Room | Median number of rooms. | 5.54 | 0.80 |
| House age | Median house age. | 31.79 | 15.98 |
| *Neighborhood attributes* | | | |
| Income (US$ in thousands) | Median household income in 2010. | 44.99 | 19.64 |
| Resident age | Median resident age. | 38.99 | 6.73 |
| Poverty | Percentage of families below a poverty line. | 16.14 | 14.90 |
| Population density | Number of people per square mile (thousands). | 1.55 | 1.53 |
| Vacant | Percentage of vacant houses. | 15.02 | 11.02 |
| Recreational | Percentage of houses used for seasonal, recreational, or occasional purposes. | 2.81 | 7.45 |
| Road (m) | Distance to primary or secondary road. | 1798.28 | 1867.78 |
| Rail (m) | Distance to the nearest active railroad track. | 6986.84 | 11,902.43 |
| School (m) | Distance to the nearest public school. | 2457.16 | 2641.52 |
| Shopping (m) | Distance to the nearest shopping center. | 6150.95 | 7806.10 |
| Hospital (m) | Distance to the nearest hospital. | 8370.56 | 7891.07 |
| Airport (m) | Distance to the nearest airport. | 11,153.00 | 7157.78 |
| *Ecosystem service attributes* | | | |
| Park (m) | Distance to the nearest public park. | 4254.13 | 4974.38 |
| Bay (m) | Distance to the nearest bay. | 7787.85 | 7991.89 |
| River (m) | Distance to the nearest river. | 6027.44 | 4151.44 |
| Stream (m) | Distance to the nearest stream. | 6506.19 | 4898.54 |
| Bayou (m) | Distance to the nearest bayou. | 8055.22 | 6312.58 |
| Water (m) | Distance to the nearest water body other than bay, river, stream, and bayou. | 995.04 | 867.39 |

### 2.3. Model Specification

A hedonic model was constructed to represent house price differentiation based on its structural, neighborhood, and ecosystem service attributes where a price of one house relative to another differs based on the level of attributes associated with each house [6,22,47,48]. When house prices are econometrically evaluated in relation to its structural, neighborhood and ecosystem service attributes, the marginal implicit price is represented by a parameter coefficient (a coefficient of attributes), whereas the relative price of a house is represented by the summation of all its marginal prices [49]. A marginal price of an ecosystem service attribute associated with a market good, such as a house, can be determined through the price a buyer paid for the property:

$$P_{Z_i}(Z_i, Z_{i-1}) = \frac{\partial P(Z)}{\partial Z_i} \tag{1}$$

where, $P$ is the price function of the vector $Z$ representing house structural, neighborhood, and ecosystem service attributes, and $i = 1, 2, \ldots, n$ is the level of different attributes describing a house.

The HPM model was first estimated using OLS as:

$$ln\,H_i = \beta_0 + \sum \beta_j S_{ij} + \sum \beta_k N_{ik} + \sum \beta_l E_{il} + \varepsilon_i \tag{2}$$

where $lnH_i$ is a natural log of an assessed value of the $i$th house in 2010 dollars, $S_{ij}$ represents $j$th house's structural attributes, $N_{ik}$ stands for $k$th neighborhood attributes, $E_{il}$ represents $l$th ecosystem service attributes, $\beta$'s are the parameter coefficients corresponding to house's structural, neighborhood, and ecosystem service attributes, and $\varepsilon_i$ is the error term.

The OLS regression model was tested for multicollinearity and heteroscedasticity. If the assumption of a constant error variance is violated, model uncertainty problem arises and in the presence of multicollinearity, estimated parameters become less efficient [50]. Thus, White's test of heteroscedasticity was used to determine if the variance of error terms was constant, whereas the Variance Inflation Factor (VIF) was used for multicollinearity test [50]. Taylor [51] suggested that price of a house may increase at a diminishing rate with some variables such as area of house and income. To capture the nonlinear relationship, Taylor [51] suggested that a functional form in HPM could have some or all independent variables transformed. In this study, house value, household income, population density and distance-related variables were logarithmically transformed, and house age was squared to represent their nonlinear nature.

A standard global modeling approach, such as OLS, might obscure the site-specific variation between a house value and its ecosystem service attributes as it cannot detect nonstationary [52]. Stationarity refers to a relationship in which house's neighborhood and ecosystem service attributes have a constant effect on the house value over space and/or time; whereas, nonstationarity indicates that the effects of the neighborhood and ecosystem service attributes vary in either or both dimensions. As house price may vary due to location, the GWR was used to address spatial nonstationarity. Steps used in developing the GWR model are represented in Figure 2.

The GWR develops a local model and estimates parameters based on neighborhood density (numbers of observations used in estimating local coefficients). The GWR approach extends the OLS regression model by producing local parameters. Following Fotheringham et al. [53], the local hedonic model utilizing the GWR was constructed as:

$$y_i = \beta_0(u_i, v_i) + \sum_k \beta_k(u_i, v_i) x_{ik} + \varepsilon_i \tag{3}$$

where $(u_i, v_i)$ are the coordinates of the $i$th point in space represented by a centroid of Census block group, $\beta_k(u_i, v_i)$ represents realization of the continuous function $\beta_k(u, v)$, and $\varepsilon_i$ is the error term. Equation (3) is the extension of Equation (2). A detailed description of the steps implemented to derive the GWR model is provided in Appendix A. An ANOVA test was conducted to test the null hypothesis that the GWR model had no improvement over the OLS model.

Local coefficients generated from the GWR model were mapped to visualize local marginal effects of ecosystem services using inverse distance weighting (IDW) interpolation method. Mennis [54] was followed in mapping t-values of local parameter estimates. A surface maps illustrating estimated coefficients and local *t*-values for ecosystem service variables were created. A t-value data layer was produced with *t*-values between −1.645 and +1.645 being masked out and t-values smaller than −1.645 or greater than +1.645 set to 100% transparency. Setting transparency for these t-value ranges to 100% allowed data stored in local coefficient layers to be unobstructed, which enabled mapping of local ecosystem service parameters significant at a 10% significance level.

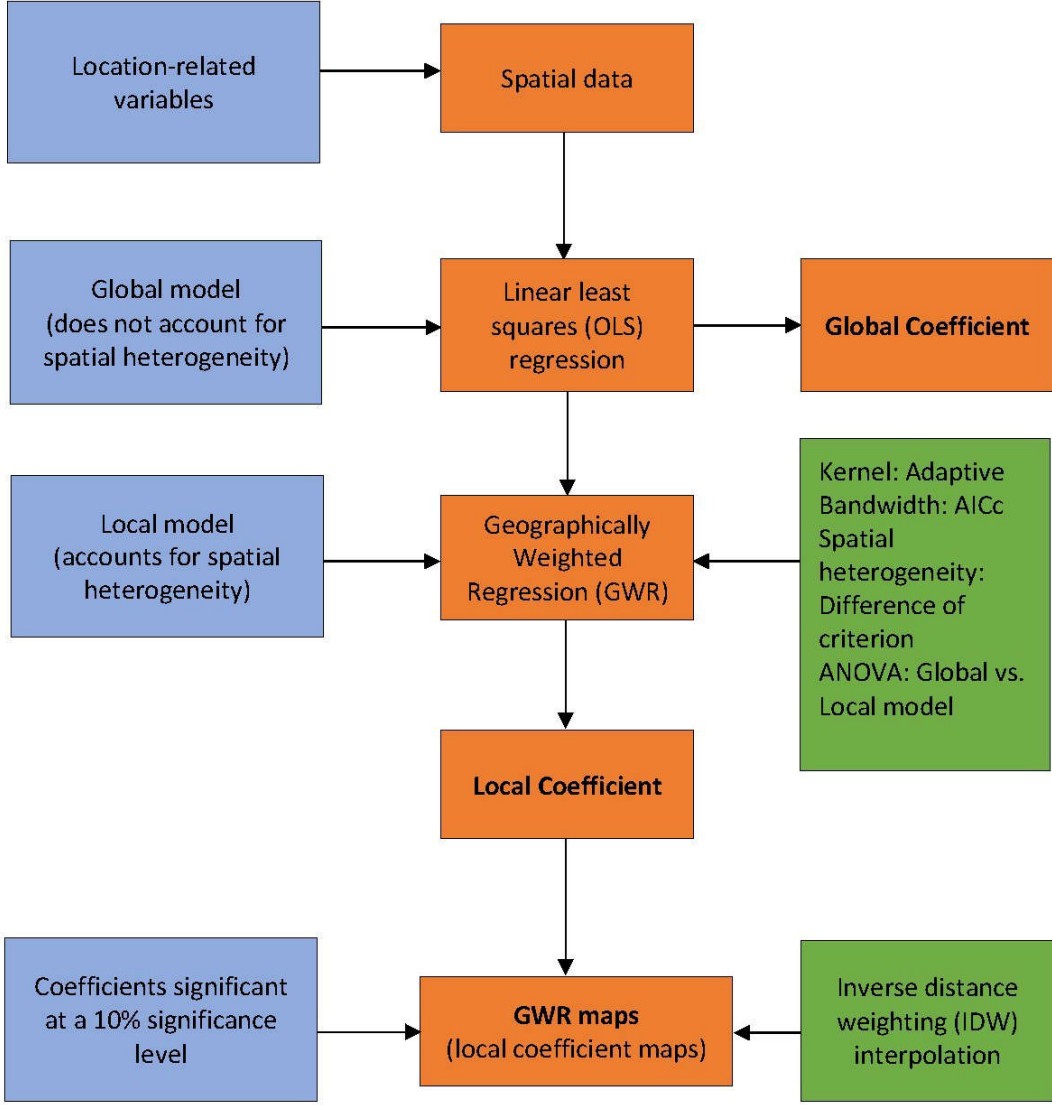

**Figure 2.** Analysis steps in extending the ordinary least squares (OLS) regression to the geographic weighted regression (GWR) model.

## 3. Results

The GWR (local) model outperformed the OLS (global) model based on ANOVA comparison (F-statistics of 2.17 and *p*-value of <0.001). Findings suggested that the GWR model was more suitable than the OLS model because it explained 57.2% of total model variation while maintaining lower RSS and AICc score. Results from geographical variability test suggested that almost all variables varied significantly suggesting they were nonstationary.

A White's test for heteroscedasticity in the OLS model (chi-square statistics = 364.09, prob > chi-square = 0.00) indicated the presence of heteroscedasticity and thus, robust standard errors were used to test for significance of model parameters. For most variables in the OLS model, VIFs were not greater than 10 except for house age and its squared term, indicating there was no multicollinearity in the model. However, these two variables were retained in the model to estimate nonlinear relationship between house age and its price [55,56]. In addition, VIF is usually higher as variables age and age-squared were correlated but does not affect the model as it does not affect the probability values of the variable [57,58].

### 3.1. Structural and Neighborhood Characteristics

House structural attributes had differing effects on house value (Table 2). OLS results suggested median number of rooms ($p < 0.001$), house age ($p = 0.027$), and house age-squared ($p = 0.005$) were statistically significant. GWR results suggested that the impacts of each of these variables were different depending on location. An increase in the median number of rooms by one was associated with a varying impact on house value ranging from 3.10% to 10.4%. Age of house was negatively associated with house value. A one-year increase in house age related to a house value decrease of 1.3% to 1.7%. However, a statistically significant positive sign of squared house age variable indicated that the house value had a nonlinear relationship with house age and the value decreased at a decreasing rate with age.

**Table 2.** Parameter estimates from the ordinary least square (OLS) global and geographically weighted regression (GWR) local models used to estimate the monetary value associated with proximity to waterfront ecosystem services in Mississippi and Alabama (U.S.), and results of the spatial variability test.

| Variables | Global Model | | Local Model | | | | | Test for Spatial Variability (Difference of Criterion) [a] |
| | Parameter coefficient | White Std. Error | Min. | Lower Quartile | Median | Upper Quartile | Max. | |
| --- | --- | --- | --- | --- | --- | --- | --- | --- |
| Intercept | 8.647 *** | 0.915 | 7.523 | 7.746 | 7.891 | 10.040 | 10.178 | 1575.438 *** |
| Rooms | 0.059 * | 0.031 | 0.031 | 0.042 | 0.049 | 0.098 | 0.104 | −35.518 *** |
| House Age | −0.014 *** | 0.003 | −0.017 | −0.016 | −0.015 | −0.014 | −0.013 | −14.978 *** |
| House Age–squared | 0.000 *** | 0.000 | 0.000 | 0.000 | 0.000 | 0.000 | 0.000 | 0.432 |
| Ln(Income) | 0.420 *** | 0.067 | 0.266 | 0.273 | 0.496 | 0.507 | 0.525 | 5057.518 *** |
| Median Age | 0.007 ** | 0.003 | −0.002 | −0.001 | 0.000 | 0.014 | 0.014 | −12.500 *** |
| Poverty | −0.001 | 0.002 | −0.002 | −0.002 | −0.001 | 0.001 | 0.001 | 1.059 |
| Ln(Pop den) | 0.000 | 0.021 | −0.031 | −0.026 | 0.014 | 0.021 | 0.036 | −20.126 *** |
| Vacant | −0.009 ** | 0.004 | −0.012 | −0.009 | −0.008 | −0.008 | −0.006 | 1.249 |
| Recreation | 0.019 *** | 0.006 | 0.009 | 0.015 | 0.022 | 0.024 | 0.033 | −0.832 * |
| Ln(Road) | 0.004 | 0.014 | −0.004 | 0.002 | 0.004 | 0.009 | 0.028 | −15.059 *** |
| Ln(Rail) | 0.013 | 0.013 | −0.021 | −0.008 | 0.011 | 0.017 | 0.028 | −15.908 *** |
| Ln(School) | 0.000 | 0.022 | −0.021 | −0.016 | 0.011 | 0.012 | 0.016 | −49.912 *** |
| Ln(Shopping centers) | −0.036 * | 0.021 | −0.051 | −0.033 | −0.029 | −0.024 | −0.013 | −51.620 *** |
| Ln(Hospital) | −0.086 *** | 0.025 | −0.157 | −0.143 | −0.137 | −0.055 | −0.032 | −29.118 *** |
| Ln(Airport) | 0.006 | 0.026 | 0.000 | 0.016 | 0.052 | 0.062 | 0.076 | −149.902 *** |
| Ln(Park) | −0.010 | 0.021 | −0.036 | −0.032 | 0.010 | 0.016 | 0.023 | −37.697 *** |
| Ln(Bay) | −0.059 *** | 0.009 | −0.064 | −0.057 | −0.054 | −0.052 | −0.047 | −9.532 *** |
| Ln(River) | −0.004 | 0.012 | −0.032 | −0.028 | −0.025 | −0.008 | −0.005 | −2.123 ** |
| Ln(Stream) | −0.030 * | 0.017 | −0.078 | −0.059 | −0.055 | −0.046 | −0.033 | −76.215 *** |
| Ln(Bayou) | 0.038 ** | 0.018 | 0.032 | 0.036 | 0.048 | 0.061 | 0.073 | −47.041 *** |
| Ln(Water) | −0.029 *** | 0.011 | −0.040 | −0.035 | −0.015 | −0.014 | −0.012 | −12.116 *** |
| $R^2$ | 0.518 | | 0.572 | | | | | |
| $R^2$ adj | 0.500 | | 0.530 | | | | | |
| AICc | 447.863 | | 427.544 | | | | | |
| CV | 0.123 | | 0.120 | | | | | |
| Residual Sum of sq. | 69.201 | | 61.425 | | | | | |
| Bandwidth (m.) | | | 533.083 | | | | | |
| N | 620 | | 620 | | | | | |

Note: The dependent variable is ln (housing value), * $p < 0.10$, ** $p < 0.05$, *** $p < 0.01$, and [a] negative value of difference criterion suggests a spatial variability.

Neighborhood characteristics had both positive and negative effects on house values (Table 2). Median household income ($p < 0.001$) and percentage of houses used for

recreational purposes (*p* = 0.001) were positively associated with house value in OLS model as well as in GWR model. A 1.0% increase in income and percentage of houses used for recreational purposes were associated with house value increases of 0.27% to 0.53% and 0.9% to 3.30%, respectively, in GWR model. However, the percentage of vacant houses was negatively related with house values. Median household age had both positive and negative relationship with house value in GWR model. Distance-related neighborhood variables, such as distance to the nearest shopping center and hospital, were negatively related to house value. With a 1.0% increase in distance to the nearest shopping center and hospital, a house value decreased by 0.013% to 0.033% and 0.032% to 0.143%, respectively, in GWR model.

### 3.2. Ecosystem Service Attributes

Proximities to different waterfront types and their ecosystem services were related to house values. Results from the OLS model showed that of six variables, four were statistically significant at the 10% or better significance level. Stationarity for each of the six variables was rejected, suggesting the monetary value of ecosystem services was spatially heterogeneous. The OLS model indicated that proximity to selected waterfront types (bay, stream, and other water body) was associated with increased house values. However, proximity to a bayou was negatively related to the house value.

The relationship of distance to different waterfronts and house values, holding all other variables constant, is plotted in Figure 3. The figure shows downward sloping relationships between house value and distance to bay, river, stream, and other water body and an upward sloping relationship with distance to a bayou. A downward sloping relationship implies that houses located nearer to waterfronts (bay, river, stream, and other water body) were assessed higher values, whereas upward slope indicated that houses located distant from waterfronts (bayou) were assessed higher values.

Marginal implicit prices for each of the significant waterfront coefficients were estimated at the mean assessed house value of $139,204 and distance of one mile (1.609 km) to the selected waterfront types (Table 3) using Equation (1). The GWR model indicated that the implicit price of distance to a bay varied across the study area (Figure 4A). The local parameter coefficients of distance to a bay estimated by the GWR model were significant throughout the study area suggesting benefits of a bay were reflected in the house value throughout the study area and the implicit price was estimated to be between $4039 and $5574 for each 1 km decrease in distance to a bay (Table 3). The GWR model suggested that the marginal implicit price, assumed to be constant across the study area in the OLS model, in fact, varied between locations and the houses were valued at a relatively higher rate on peripheries of urban centers than other areas (Figure 4A).

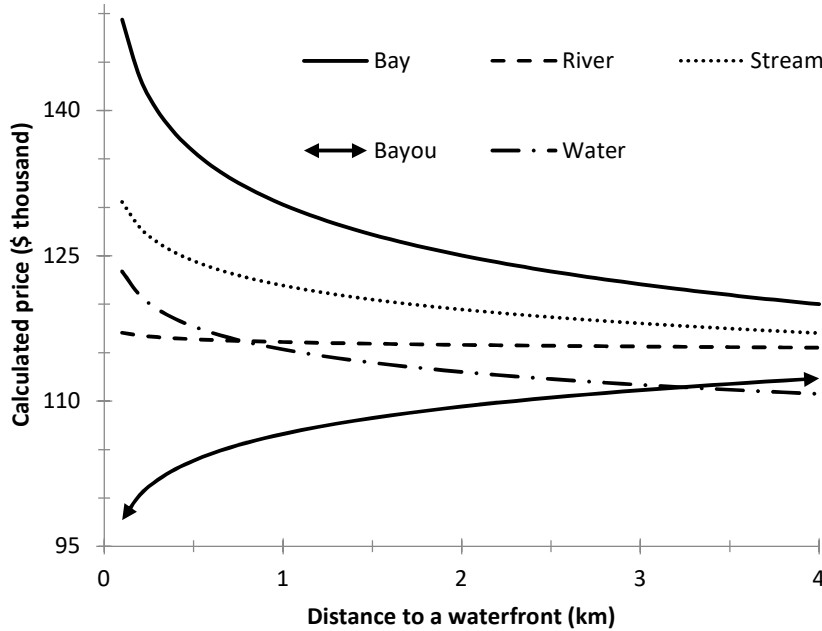

**Figure 3.** Relationships between assessed house value and proximity to waterfront ecosystem services.

**Table 3.** Estimated marginal implicit prices per 1 km decrease in distance to waterfront ecosystem services.

| Waterfront Type | Marginal Implicit Price per km | |
| --- | --- | --- |
| | OLS | GWR * |
| Nearest bay | $5133 | $4039 to $5574 |
| Nearest stream | $2566 | $2898 to $6773 |
| Nearest river | NS | $2455 to $2802 |
| Nearest bayou | −$3267 | −$3608 to −$6343 |
| Nearest other water body | $2534 | $2287 to $3432 |

Note: NS represents a statistically insignificant coefficient. * Only significant GWR estimates were translated to US$ values.

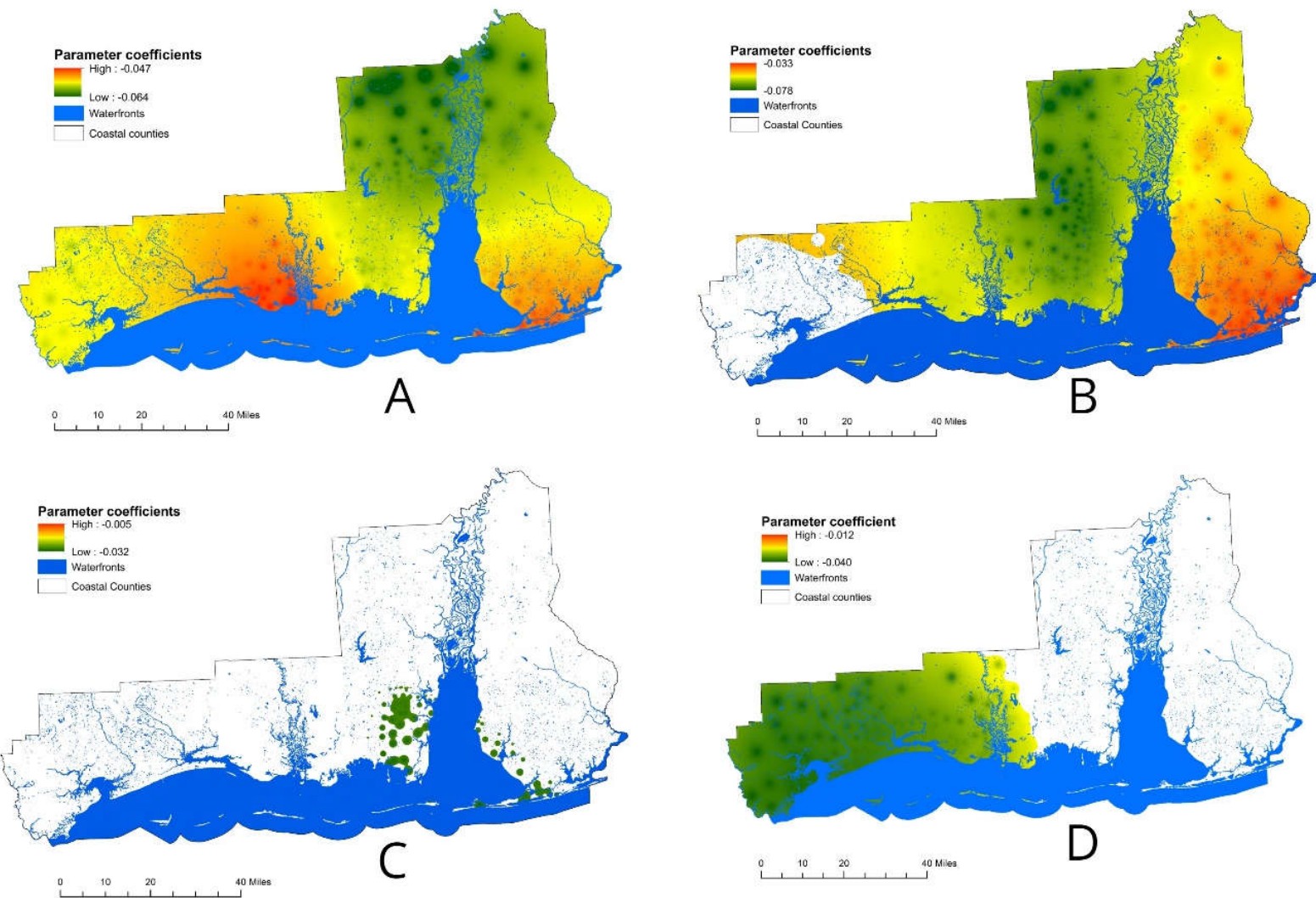

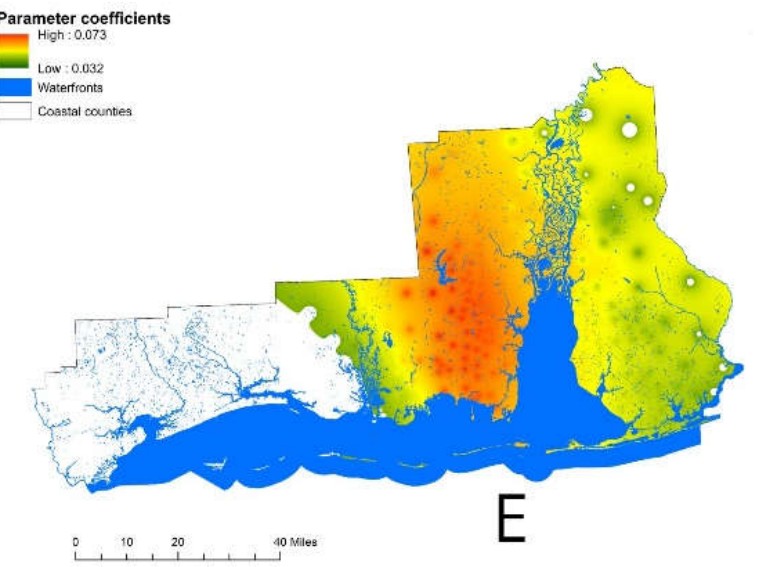

**Figure 4.** Local parameter coefficients of proximity to waterfront ecosystem services estimated from GWR model that were statistically significant at 10% or a better significance level: (**A**) bay, (**B**) stream, (**C**) river, (**D**) other water body, and (**E**) bayou.

Proximity to the nearest stream was also valued positively by coastal residents. The GWR model demonstrated variability of the implicit price which ranged between $2898 and $6773 for 1 km decrease in distance to a stream. Figure 4B illustrates that local parameters for a stream were not statistically significant throughout the study area (indicated by white color) suggesting distance to stream did not have any effect on house values in some locations. Proximity to a stream was more valued in Mobile County than Baldwin County, whereas it was not associated with house values in Hancock County and some regions of Harrison County. Proximity to a river was not associated with house values in the OLS model; however, in the GWR model, some parameters in the southern region of the study area were related to house values (significant at 10% or a better significance level) (Figure 4C). People living near Dog River, East and West Fowl Rivers, and Little River valued proximity to a river, whereas in other areas proximity to a river did not have an association with house values. The marginal implicit price was estimated between $2455 and $2802 per 1 km decrease in distance. Similarly, Figure 4D illustrated the spatial variability due to distance to other water body. The estimated implicit price was between $2287 and $3432 per 1 km decrease in distance (GWR). Proximity to other water bodies was significant in coastal counties of Mississippi but not in Alabama. Findings implied that the proximity to other water bodies was valued by residents where streams and quality bays were relatively scarce, such as in Hancock and Harrison Counties.

An examination of Figure 4E indicated spatial variability in the effect of distance to a bayou on house values. The estimated marginal implicit price from the GWR model was between −$3608 and −$6343 per 1 km decrease in distance suggesting that proximity to a bayou was negatively associated with house values. This negative relationship between house values and proximity to a bayou occurred in almost all parts of the study area in Alabama; however, it was mostly statistically insignificant in Mississippi.

## 4. Discussion

Traditional regression models such as OLS can mask important local variation in model parameters [59]. In most previous studies, the relationship between residential property value and proximity to waterfronts represented an average measure by assuming the relationship did not change across the landscape. However, this study revealed that this assumption is not necessarily true. Ignoring the notion that specific sites within the same study area might differ from each other with regard to availability of ecosystem services and their impact on property values might lead to incorrect generalizations about the entire study area [60]. Inferences based on results from a global OLS model, where nonstationarity is present, might not be sufficient in a local policy setting [39].

This study suggested there was spatial variation and the global (OLS) model did not capture the geographical variations in house values across the study area and provided ambiguous coefficients reflecting a net effect of proximity to waterfront ecosystem services that may not be appropriate for all areas. For example, proximity to stream and other water bodies corresponded to an increase in house value by 0.03%. However, GWR estimated that effect of proximity to a stream ranged from 0.03% to 0.07% reflecting an increase of $2800 to $6800 in house value for a 1 km decrease in distance to the nearest stream. Similarly, other water body estimates ranged from 0.01% to 0.04% reflecting a $1000 to $3000 increase in house value for a 1 km decrease in distance. These findings suggest that waterfront ecosystem services were valued differently not only for different waterfront types but also for different locations. GWR maps illustrated that OLS results were suboptimal as a predicted response varied across the study area. For example, proximity to a bay was assigned higher values in urban peripheries than other regions. Thus, findings from GWR provide some guidance as to how policy solutions need to be spatially varied across geographic area. Information on spatial variation of waterfront ecosystem service monetary value, in conjunction with information on their ecological

importance, can be used to identify areas requiring preservation and those more suitable for development, and implementing local policies, such as zoning regulations, that will balance future changes in urban land use. Conservation easements, for example, can be modified to fit specific site conditions and ecosystem availability based on findings from this research without the necessity to adopt the same policy for entire the study area. As residents preferred to live in proximity to most waterfront types, planners can use this information to formulate site-specific conservation strategies in allocating budgets for both waterfront ecosystem service conservation and development in city planning. On the other hand, bayous, were less desired by residents; therefore, surrounding areas could be targeted for preservation.

Proximity to a bay was associated with the greatest monetary value, followed by streams and other water bodies, which suggested that large-size waterfronts were appreciated more than smaller ones. The possible reason might be that larger-size waterfronts provide more ecosystem services and recreational opportunities, such as boating, than smaller-size waterfronts. Mahan et al. [30] found a similar trend where a $24.39 increase in a house value was associated with one-acre increase in adjacent wetland size. Similarly, Bolitzer and Netusin [29], Habb and McConnell [61], and Poudyal et al. [55] indicated an increase in residential property values with an increase in size of open space. In addition, ecosystem services of bays might have been captured better than other types of waterfronts, such as rivers and streams, because the study area is in the coastal region where access to bay areas is fairly easy. Thus, coastal residents preferred to live closer to waterfronts, and size and types of waterfronts seemed to matter in their decisions. The information on waterfront size and type preference can be used to promote availability, improve access to, and increase provision of ecosystem services associated with certain waterfront types. This approach can also help increase property tax revenues that can be used to improve urban infrastructure and services and enhance availability of ecosystem services in waterfront areas preferred by residents.

Proximity to rivers, streams and other water bodies such as lakes, small ponds, and reservoirs were also valued positively. This result was consistent with previous studies by Mahan et al. [30] and Sander and Polasky [62] who estimated 0.004% to 0.01% increases in house value with a 1% increase in proximity to the nearest stream. Bowman et al. [63] estimated that adjacency to a stream increased a house value by 9.60%. Doss and Taff [64] found that with a 10-m increase in proximity to a lake, a residential property value increased by $187.92, whereas Sander and Polasky [62] reported a 0.01% increase in residential house sale price with a 1-percent increase in proximity to the nearest lake. Similarly, Luttik [65] estimated that house prices increased by 10% with access to a water view. Bayou, however, had a positive coefficient sign suggesting that residents assessed higher value to houses located away from this waterfront type. A possible reason might be that bayous are less appealing because of their bio-physical characteristics and provide habitat for many wild animals, such as alligators, and thus residents might consider living in proximity to bayous as unsafe. Another reason might be residents preferred diverse recreational opportunities associated with bays and stream than bayous. Bayous provide more of nonuse value compared to other waterfront types such as bays, streams, and rivers which cannot be measured by hedonic model employed in this study.

The study used data at the Census block group level that lacked many house structural attributes such as number bathrooms; presence of garage, fireplace, porch, and central air conditioning system; area of a property; and quality of a house, which are crucial in HPM analysis. As other relevant information was missing, the analysis may potentially have suffered from inefficiency. However, all available variables were used in the analysis to control the variation across housing stock caused by structural attributes. In addition, the data available from the Census are presented as either median or average assessed house values for a Census block group. The data aggregation to a Census block group may have impacted the reliability of estimated marginal implicit prices compared to estimates derived based on individual house-level information. However, Shultz and

King [47] suggested that aggregating data by a block group level is preferable over the block or tract level. Despite some disadvantages of Census data, overall quality of waterfront ecosystem service values as embodied in assessed house values was well represented because estimate coefficient signs and significance were consistent with Dahal et al. [24] who performed a similar study in the same location and time frame using detailed information from MLS data. However, it is recommended that additional HPM studies should be conducted to evaluate consistency of their estimates with those derived based on detailed information obtained from MLS or tax assessors. Despite potential limitations, monetary assessment of waterfront ecosystem services based on Census block group data would be helpful in situations where MLS or tax assessors' information is either unavailable or difficult to acquire and thus not practical for local leaders who could use the information in their land-use decisions.

While the study demonstrated the usefulness of GWR in addressing spatial nonstationarity considering all variables were spatially heterogeneous, in the real world all independent variables might not necessarily vary across spatial locations [65]. For example, ecosystem service attributes expressed as proximity to waterfronts may be valued differently in different locations, whereas some structural variables, such as a number of bedrooms, may be valued equally in different locations. Therefore, future studies should consider using a mixed geographically weighted regression (MGWR) approach where some of the variables are allowed to be global (stationary) and others to be local (nonstationary). The MGWR approach can explore spatial variation of some independent variables as well as allow global effect of the stationary variables. The mixed approach would be more flexible and parsimonious than GWR model and would improve model efficiency [66,67].

This study did not measure the actual size of waterfronts and made inferences based on observations that bays were larger in size than streams and other water body types. Information on actual size of waterfronts would aid in determining whether the waterfront outputs are amenities or disamenities to coastal residents. For instance, waterfronts that are relatively large in size allow diverse recreational opportunities and homeowners may be easily attracted towards them while waterfronts that are smaller in size might be considered merely a pond in buyer's eyes. The study did not account for the impact of waterfront view or accessibility. Future studies should consider waterfront view as one of the potential ecosystem services as unobstructed waterfront view may serve as superior selling attribute to attract potential buyers. The study used single-year data, 2010 Census data, to estimate monetary values of proximity to waterfront ecosystem services. Utilizing housing data from an extended time period would have helped to control for market fluctuations due to many exogenous events by averaging their effects.

## 5. Conclusions

The study determined how monetary value of proximity to waterfront ecosystem services varied across a geographic area and by a waterfront type, and demonstrated that the aggregated information as available in the U.S. Census can be used to estimate a monetary value of ecosystem services associated with waterfronts. The information on resident preferences related to open space areas, such as waterfronts, will be helpful in identifying areas suitable for natural resource conservation versus development and adopting local land-use policies that will facilitate this process. For example, bays were highly preferred in urban centers especially in Mobile and Baldwin Counties and as an increasing number of people have been choosing to live nearby such areas, this might create land use pressure and potential loss of waterfront ecosystem services. Therefore, policies regarding preservation of waterfronts should be focused on such areas to maintain local communities' interests and ecological values. Findings from this study provide crucial information for city planners to promote healthy environments where local economies can benefit from increased property tax revenues associated with different waterfronts. Planners can use this information to develop guidelines for

waterfront preservation and urban development by maintaining visually appealing and easily accessible waterfronts in close vicinity to urban settings.

Results have several implications. Estimates from the GWR model indicated that the marginal price of proximity to waterfronts was not constant throughout the study area, suggesting that a general conservation/development policy might not be appropriate for a local/regional setting as there was a substantial spatial variation in monetary value of proximity to waterfront ecosystem services. The information on waterfront value variation can be used to educate decision-makers about the relative value of waterfronts and associated ecosystem services and facilitate new ideas on how to better adopt location-specific land-use management that fits local characteristics and is preferable by local communities. Aggregated Census data can be used to produce quick and first-hand estimates that local decision-makers can use to make more informed land-use decisions when detailed MLS data are not available, difficult to acquire, or costly. For example, it can be utilized to develop a Web site or a mobile application that local decision-makers can use to acquire needed data quickly and easily make their decisions regarding land-use management in a specific area. A quick access to data and efficient technique for monetary valuation of waterfront ecosystem services can serve as an important decision tool in designing urban waterfront policies that are better aligned with local settings and lead to higher economic activity while preserving their ecosystem services.

**Author Contributions:** Conceptualization, R.P.D., R.K.G., J.S.G., I.A.M. and D.R.P.; methodology, R.P.D., R.K.G. and J.S.G.; data curation, R.P.D.; software, R.P.D.; investigation, R.P.D., R.K.G. and J.S.G.; writing—original draft preparation, R.P.D.; writing—review and editing, R.P.D., R.K.G., J.S.G., I.A.M. and D.R.P.; validation, R.K.G. and J.S.G.; funding acquisition, J.S.G. and R.K.G.; project administration, R.K.G. and J.S.G.; supervision, R.K.G., J.S.G., I.A.M. and D.R.P. All authors have read and agreed to the published version of the manuscript.

**Funding:** This publication was supported by the U.S. Department of Commerce's National Oceanic and Atmospheric Administration under NOAA Award NA14OAR4170098, the Mississippi-Alabama Sea Grant Consortium and Mississippi State University. The views expressed herein do not nec-essarily reflect the views of any of these organizations.

**Institutional Review Board Statement:** Not applicable.

**Informed Consent Statement:** Not applicable.

**Data Availability Statement:** Not applicable.

**Acknowledgements:** This publication was supported by the U.S. Department of Commerce's National Oceanic and Atmospheric Administration under NOAA Award NA14OAR4170098, the Mississippi-Alabama Sea Grant Consortium and Mississippi State University. The views expressed herein do not necessarily reflect the views of any of these organizations.

**Conflicts of Interest:** The authors declare no conflict of interest.

## Appendix A

In the GWR model, house structural, neighborhood and ecosystem service attributes located closer to a house $i$ have a greater impact in the estimation of a local parameter, $\hat{\beta}_k(u_i,v_i)$, than for a house located away from location $i$. Thus, the model [from Equation (3)] can be further represented as:

$$\hat{\beta}_k(u_i,v_i) = (X'W(u_i,v_i)X)^{-1}X'W(u_i,v_i)y \qquad (A1)$$

where $\hat{\beta}_k(u_i,v_i)$ is an estimate of $\beta(u_i,v_i)$ and $W(u_i,v_i)$ is an $n \times n$ matrix with off-diagonal elements as zero and diagonal elements representing the geographical weight of each location $(u_i,v_i)$ as a function of distance from one location to other location for which observations were collected:

$$= \begin{bmatrix} \beta_0(u_1,v_1) & \beta_1(u_1,v_1) & \cdots & \beta_k(u_1,v_1) \\ \beta_0(u_2,v_2) & \beta_1(u_2,v_2) & \cdots & \beta_k(u_2,v_2) \\ \vdots & \vdots & & \vdots \\ \beta_0(u_n,v_n) & \beta_1(u_n,v_n) & \cdots & \beta_k(u_n,v_n) \end{bmatrix} \tag{A2}$$

The parameters in each row of the above matrix were estimated by:

$$\hat{\beta}(i) = (X'W(i)X)^{-1}X'W(i)Y \tag{A3}$$

The spatial weighting matrix was represented by:

$$W(i) = \begin{bmatrix} w_{i1} & 0 & \cdots & 0 \\ 0 & w_{i2} & \cdots & 0 \\ \vdots & \vdots & \ddots & \vdots \\ 0 & 0 & \cdots & w_{in} \end{bmatrix} \tag{A4}$$

Equation (A4) represents the weighted least squares estimator where the weight varies according to the location $i$. $W(i)$ was computed with a bi-square kernel function. The function provides a continuous, near-Gaussian weighting function up to distance $b$ from the regression point and then zero weight to any data point beyond $b$ [53]. While the kernel can either be a fixed or adaptive distance, an adaptive distance function was selected because the spatial context of the study area was relatively large and geographic feature (centroid of Census block group) distribution was irregular [68]:

$$w_{ij} = \begin{cases} \left[ 1 - \left( \dfrac{d_{ij}}{b} \right)^2 \right]^2, & d_{ij} < b \\ 0, & d_{ij} > b \end{cases} \tag{A5}$$

where $w_{ij}$ is the weight function, $j$ is a specific data point in space, $d_{ij}$ is the Euclidean distance between locations $i$ and $j$, and $b$ is an adaptive bandwidth size defined as the $k$th nearest neighbor distance. The $w_{ij}$ equals one if $j$ is one of the *Kths* nearest neighbors of $i$. The value of $K$ is the number of observations that are included in the GWR model. Weights for all data points beyond the $K$th become zero. The selection of $d$ in Equation (A5) is important because as $d$ increases, the local model will be closer to global model and as $d$ decreases, parameter estimates progressively depend on observations in close proximity to $i$ and thus increase variance [53]. A geographically weighted regression model was estimated using GWR4 software.

This study used the Golden Section Search routine to identify a bandwidth minimizing the Akaike Information Criterion (AIC) score [69]. The AIC has an advantage of being more general and accounts for model parsimony [53]. As the classic AIC selects smaller bandwidths which are likely to be undersmoothed, a corrected AIC (AICc) was used as a selection criterion [70]. A model with a lower AICc is considered to be the better model because the smaller the AIC score, the closer will it be to the true model [53]. The AICc was defined as:

$$AIC_c = 2nlog_e(\hat{\sigma}) + nlog_e(2\pi) + n\left\{ \frac{n + tr(s)}{n - 2 - tr(s)} \right\} \tag{A6}$$

where $n$ is the sample size, $\hat{\sigma}$ is the estimated standard deviation of the error term, and $tr(s)$ represents the trace of the hat matrix (mapping $\hat{y}$ onto $y$) and is a function of the bandwidth:

$$\hat{y} = sy \tag{A7}$$

Local standard errors were based on the variance matrix and expressed as:

$$\hat{y} = sy \tag{A8}$$

where *C* represents the variance-covariance matrix and was defined as:

$$C = (X'W(u_i,v_i)X)^{-1}X'W(u_i,v_i) \tag{A9}$$

After computing the variance of each parameter estimate, standard errors were obtained as follows:

$$SE(\hat{\beta}_i) = sqrt[Var(\hat{\beta}_i)] \tag{A10}$$

The weighting matrix addressed the spatial heterogeneity, whereas the statistical significance of each local parameter estimate obtained from the GWR model was evaluated using a *t*-test. To test for spatial heterogeneity of the GWR model, a spatial variability was examined. The variability for each parameter coefficient was tested by model comparison between the fitted GWR (estimated *y* values computed by GWR) and a model in which only the *k*th coefficient was fixed while others were kept as they were in the fitted GWR.

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
