# Peer review of "Geospatial Heterogeneity in Monetary Value of Proximity to Waterfront Ecosystem Services in the Gulf of Mexico"

_water, doi:10.3390/w13172401_

Round 1

Reviewer 1 Report

The work presented in the manuscript is very interesting and valuable in the field. However, there are a few minor recommendations/suggestions:

  • In the introduction section it is presented the situation of the population, but omitted to be revealed the later/current situation.
  • Line 139 – I am not sure everyone knows what type of source Zillow is, a brief description would be helpful
  • Table 1 can be placed in line 187 were introduced in text or if decided to be left at the end of section 2 than an explanation should be added right before it.
  • Line 202 (equation 2) – please check the index for beta and E to correspond to the explanation in text
  • Please provide reference to the statement in lines 218-220 about the nonstationarity
  • In the results section, the explanations from lines 310 to 314 can be omitted since it continues with a clear notice on the fact that GWR is more suitable than OLS, and all those indicator scan be read from Table 2.
  • Discussion section should be renumbered 4 and Conclusions 5
  • please review the editing instructions, there are some minor things to correct such as: in abstract there is an empty line, title of Figure 1 should be right below the figure, spacing in reference list
  • And finally just a curiosity: Are you planning to redone the study on the new census? What changes do you expect to find?

Author Response

Comment: The work presented in the manuscript is very interesting and valuable in the field. However, there are a few minor recommendations/suggestions:

Reply: Thank you.

Comment: In the introduction section it is presented the situation of the population but omitted to be revealed the later/current situation.

Reply: Thank you for your comments. We have revised the population status description and included information describing a current situation.

“…the U.S. coastal population, including Atlantic, Gulf of Mexico, and Pacific, grew by 40 million between 1960 and 2008, representing an increase of 84% [13]. During the same time period, the U.S. population in the coastal counties along the Gulf of Mexico increased by 150%, whereas in five coastal counties in Mississippi and Alabama, including Hancock, Harrison, Jackson, Mobile, and Baldwin, population increased by over 65% taking into account a population decline after numerous hurricanes such as Katrina [13]. Similarly, between 2010 and 2019, population in these five counties increased by another 8% overall, of which Baldwin County experienced the largest increase of over 20% [14].”

Comment: Line 139 – I am not sure everyone knows what type of source Zillow is, a brief description would be helpful

Reply: We have now added brief statement to read, “… (one of the leading online real estate and rental marketplaces reporting information related to median house values, inventory, and other parameters)”

Comment: Table 1 can be placed in line 187 were introduced in text or if decided to be left at the end of section 2 than an explanation should be added right before it.

Reply: Thank you for the suggestion. We have moved Table 1 to data collection section.

Comment: Line 202 (equation 2) – please check the index for beta and E to correspond to the explanation in text

Reply: we have corrected the issue.

Comment: Please provide reference to the statement in lines 218-220 about the non-stationarity

Reply: Reference has been added.

Comment: In the results section, the explanations from lines 310 to 314 can be omitted since it continues with a clear notice on the fact that GWR is more suitable than OLS, and all those indicator scans be read from Table 2.

Reply: Thank you. We have removed those lines.

Comment: Discussion section should be renumbered 4 and Conclusions 5

Reply: Thank you for catching that. While submitting the MS-Word document of our manuscript, the system automatically changed some of the formatting resulting in some minor errors such as empty line in the abstract, numbering, location of figure titles, etc. We have corrected them and will check again during the article proofing.

Comment: please review the editing instructions, there are some minor things to correct such as: in abstract there is an empty line, title of Figure 1 should be right below the figure, spacing in reference list

Reply: Thank you. Formatting issues in the manuscript have been corrected. Some of these issues emerged after automatic conversion to PDF.

Comment: And finally, just a curiosity: Are you planning to redone the study on the new census? What changes do you expect to find?

Reply: 2020 Census data is not released yet. 2010 Census was the best available data when the project was started. We expect, if newer data was used or will be used in future, results will be similar to what we found. A survey done by Dahal et al. (2019) in the Gulf of Mexico regarding residents’ attitudes towards open space and working waterfront revealed that respondents valued ecosystem services and over 95% of residents believed that ecosystem services were important and should be preserved in future development decisions. Residents are now more aware and educated about ecosystem services and thus, we believe that marginal price of proximity to different waterfront types can potentially be different (or even higher) suggesting its importance to coastal communities.

Reviewer 2 Report

Dear Authors

The manuscript entitled "Geospatial Heterogeneity in Monetary Value of Proximity to Waterfront Ecosystem Services in the Gulf of Mexico" estimated the monetary value of distance to different waterfront types in coastal counties of Mississippi and Alabama (USA) using a geographically weighted regression (GWR) approach as an extension to a traditional hedonic pricing method (HPM). 

 Findings and suggestions from this study can be used in developing healthy living environments where local economy can benefit from increased property tax revenues associated with waterfronts and their ecosystem services. 

The manuscript is well planned and presented, results are adequate and discussed nicely. The conclusion looks very big which may be summarized to 2 small paragraphs.

Thank you

Author Response

Comment: The manuscript entitled "Geospatial Heterogeneity in Monetary Value of Proximity to Waterfront Ecosystem Services in the Gulf of Mexico" estimated the monetary value of distance to different waterfront types in coastal counties of Mississippi and Alabama (USA) using a geographically weighted regression (GWR) approach as an extension to a traditional hedonic pricing method (HPM). Findings and suggestions from this study can be used in developing healthy living environments where local economy can benefit from increased property tax revenues associated with waterfronts and their ecosystem services. The manuscript is well planned and presented, results are adequate and discussed nicely. The conclusion looks very big which may be summarized to 2 small paragraphs

Reply: Thank you. We have shortened the conclusion section and included only a general summary in the first paragraph and policy implication in the second paragraph.

Reviewer 3 Report

The value of proximity of house to waterfront ecosystem service in the Gulf of Mexico was explored in this study. The proper evaluation of non-market ecosystem service by waterfront is important for policy maker to plan the area development near the waterfront.
This study provided a useful procedure, results and suggestions not only for academic researcher but also for policy makers or city developers.

In my opinion, some parts in materials and methods should be in introduction part as they contain background information other than methods.

P.4, L134-146

The contents of this paragraph seem to be more introductory. So these should be in introduction part in a concise form.

P.5 L189- P.7 305

Also in 2.3. Model Specification part, there seems to be a bit too much information was shown, some of which were maybe more like supplementary. 

In the results part, P.9 L 333 - P.10 L.356

The results shown in this part had better to be shown in a form of table or figure at least in appendix or in supplementary material. Then it would be more understandable and informative.

Also, the discussion part, which is currently a part of "3. Results", should be an independent section from the results section with subheads. 

Author Response

Comment: The value of proximity of house to waterfront ecosystem service in the Gulf of Mexico was explored in this study. The proper evaluation of non-market ecosystem service by waterfront is important for policy maker to plan the area development near the waterfront.
This study provided a useful procedure, results and suggestions not only for academic researcher but also for policy makers or city developers.

 Reply: Thank you.

Comment: In my opinion, some parts in materials and methods should be in introduction part as they contain background information other than methods.

P.4, L134-146. The contents of this paragraph seem to be more introductory. So these should be in introduction part in a concise form.

Reply: Thank you. The paragraph has been moved to the introduction section.

Comment: P.5 L189- P.7 305. Also, in 2.3. Model Specification part, there seems to be a bit too much information was shown, some of which were maybe more like supplementary. 

Reply: Thank you. We have added GWR flowchart (Figure 2) for those readers who would like to know the steps without going through detailed methodology. The flow chart provides quick high-level information on what steps were followed during analysis. A substantial portion of a detailed methodology description has been moved to the appendix.

Comment: In the results part, P.9 L 333 - P.10 L.356. The results shown in this part had better to be shown in a form of table or figure at least in appendix or in supplementary material. Then it would be more understandable and informative.

 Reply: Thank you for catching that. In fact, results are discussed based on Table 2. We have added reference in this section as Table 2.

Comment: Also, the discussion part, which is currently a part of "3. Results", should be an independent section from the results section with subheads. 

Reply: While submitting the MS-Word format of our manuscript, the system automatically changed word file to PDF changing some of the formatting. As a result, the discussion section was numbered as 3.3 as subsection of results instead of separate section as 4. We have corrected it and will check again during the article proofing.

Reviewer 4 Report

Dear Authors,

The topic of the article is interesting and within the scope of Water. The manuscript has certainly the potential to improve. In my humble opinion, if the manuscript is thoroughly revised and reorganized, it can make a fine publication. To help improve the quality of this manuscript, I have added more comments below:

General Comments:

  1. The abstract should be revised and clearly define the problem statement and solution.
  2. Why there is a gap in the abstract section?
  3. Kindly extend the Introduction part by providing recent references and should explain why this study is important than others? research gap for this study. 
  4. It is better to draw a methodology figure so that reader can read it clearly. 
  5.  It is better to provide policy implications in the conclusion section. 
  6. Please enter the appropriate SI unit of measurement next to the label. Check the full text.
  7. Correct the "References" section in accordance with the "Instructions for Authors".

Best regards, 

Author Response

Comment: The topic of the article is interesting and within the scope of Water. The manuscript has certainly the potential to improve. In my humble opinion, if the manuscript is thoroughly revised and reorganized, it can make a fine publication. To help improve the quality of this manuscript, I have added more comments below:

Reply: Thank you.

Comment: The abstract should be revised and clearly define the problem statement and solution.

Reply: Thank you. Open spaces, such as waterfront areas, are now well recognized and have been accounted for in many land-use decisions making process. However, it is often not clear how values of ecosystem services associated with waterfronts vary across geographical areas which prevents decision makers from developing site-specific conservation plans. Thus, this study estimated geographical variability of values associated with different types waterfront open space so that the information can be used in developing local policies. In addition, data typically required to conduct such studies, such as MLS data, are often difficult to acquire and sometimes costly. We used publicly available Census data, tested feasibility of its use, and determined that the estimates were consistent and will be helpful in situations where MLS or tax assessor’s information is unavailable and difficult to acquire. Revised abstract with problem and solutions are mentioned in line 6-15.

“… it is not clear how values of waterfront ecosystem services vary across geographical areas which prevents development and adoption of site-specific natural resource conservation plans and suitable long-term land management strategies. This study estimated the monetary value of distance to different waterfront types in coastal counties of Mississippi and Alabama (U.S.) using a geographically weighted regression (GWR) approach as an extension to a traditional hedonic pricing method (HPM). In addition, the study utilized publicly available data from the U.S. Census Bureau instead of certified rolls of county property assessors and Multiple Listing Service (MLS) data which can be costly and difficult to obtain.

Comment: Why there is a gap in the abstract section?

Reply: The gap was generated by the system when the MS-Word document, that we initially submitted, was changed to PDF. There are also other editing issues that resulted from the same conversion process. We have corrected them and will again check for these issues during the article proofing.  

Comment: Kindly extend the Introduction part by providing recent references and should explain why this study is important than others? research gap for this study. 

Reply: Thank you. We extended the introduction section and added recent references to explain why the spatial model (GWR) is more applicable than the global (OLS) model (pages 6-7; lines 110-126). As OLS is the traditional measure of hedonic model, we used OLS as our first step model and extend it to the spatial model (GWR). Most of the HPM studies are still using OLS model to estimate nonmarket values of ecosystem services. Findings from OLS model may provide a useful benchmark for making general statements but may not reflect monetary values of environmental goods in specific geographic locations which was the focus of our study. Therefore, we selected the GWR model to account for the nonstationarity that provides crucial site-specific information to planners, natural resource managers, and other decision makers.

Many previous valuation studies have considered the impact of waterfront proximity on house value; however, they did not account for the effect of spatial variation in the impact of waterfront ecosystem services on house values (e.g. [27–35]. For example, Cohen et al. [36] demonstrated that the ordinary least squares (OLS) regression method did not indicate the relationship between house price and proximity to waterbody; however, the geographically weighted regression (GWR) method suggested that a 1-percent increase in distance to a waterbody was related to a house price decrease of 0.027%. Similarly, Pandit et al. [37] and Tapsuwan et al. [38] suggested the use of spatial models to account for the impact of location on house prices because spatial dependencies can result in biased and inconsistent estimates when they are derived from stationary models such as the OLS. Information obtained from the OLS model can only provide average estimates that can serve as a useful benchmark for making general statements; however, such estimates might not accurately reflect monetary values of ecosystem services associated with specific sites [39]. Thus, there is a need to account for spatial heterogeneity when quantifying a monetary value of ecosystem services associated with waterfronts to better understand how these attributes are related to house values and facilitate an improved planning for open space conservation and development.

Comment: It is better to draw a methodology figure so that reader can read it clearly. 

Reply: We appreciate the comment. To assist reader in following methodology, we have added a methodology figure (Figure 2]. 

Comment: I It is better to provide policy implications in the conclusion section. 

Reply: Thank you. We have added policy implications in the second paragraph of the conclusion section.

“Results have several implications. Estimates from the GWR model indicated that the marginal price of proximity to waterfronts was not constant throughout the study area, suggesting that a general conservation/development policy might not be appropriate for a local/regional setting as there was a substantial spatial variation in monetary value of proximity to waterfront ecosystem services. The information on waterfront value variation can be used to educate decision makers about the relative value of waterfronts and associated ecosystem services, and facilitate new ideas on how to better adopt location-specific land-use management that fits local characteristics and is preferable by local communities. Aggregated Census data can be used to produce quick and first-hand estimates that local decision makers can use to make more informed land-use decisions when detailed MLS data are not available, difficult to acquire, or costly. For example, it can be utilized to develop a Web site or a mobile application that local decision makers can use to acquire needed data quickly and easily make their decisions regarding land-use management in a specific area. A quick access to data and efficient technique for monetary valuation of waterfront ecosystem services can serve as important decision tool in designing urban waterfront policies that are better aligned with local settings and lead to higher economic activity while preserving their ecosystem services.” 

Comment: Please enter the appropriate SI unit of measurement next to the label. Check the full text.

Reply: We have added unit of measurement next to the label.  

Comment: Correct the "References" section in accordance with the "Instructions for Authors".

Reply: References have been revised according to the journal instructions.

Round 2

Reviewer 1 Report

The manuscript was improved according to the recommendations.

Thank you for the additional explanation related to the next census and good luck in future research work.

Reviewer 4 Report

Agree with the revision of the authors. No further comments.